# Circulating Blood-Brain Barrier Proteins for Differentiating Ischaemic Stroke Patients from Stroke Mimics

**DOI:** 10.3390/biom14111344

**Published:** 2024-10-22

**Authors:** Pragati Kakkar, Meaad Almusined, Tarun Kakkar, Theresa Munyombwe, Linetty Makawa, Kirti Kain, Ahamad Hassan, Sikha Saha

**Affiliations:** 1Leeds Institute of Cardiovascular and Metabolic Medicine, University of Leeds, Leeds LS2 9JT, UK; p.kakkar@leeds.ac.uk (P.K.); malmusined@ksu.edu.sa (M.A.); t.munyombwe@leeds.ac.uk (T.M.); 2School of Chemical and Process Engineering, University of Leeds, Leeds LS2 9JT, UK; t.kakkar@leeds.ac.uk; 3Leeds Institute for Data Analytics, University of Leeds, Leeds LS2 9JT, UK; 4Leeds Teaching Hospitals NHS Trust, Leeds LS1 3EX, UK; linetty.makawa@nhs.net (L.M.); ahamad.hassan@nhs.net (A.H.); 5NHS England-North-East and Yorkshire, Professional Standards, Leeds LS2 7UE, UK; k.kain@nhs.net

**Keywords:** ischaemic stroke, blood-brain barrier, NIHSS, Glial Fibrillary Acidic Protein, biomarker

## Abstract

Background: Stroke is one of the leading causes of death and disability worldwide. The diagnosis of stroke remains largely clinical, yet widely used stroke scoring systems and brain imaging do not satisfactorily allow the distinction of ischaemic stroke (IS) patients from stroke mimics (SMs). Blood biomarkers are promising tools that could facilitate clinical triage. Methods: This study recruited 66 patients with IS and 24 SMs. The levels of Glial fibrillary acidic protein (GFAP), Neuron-specific enolase (NSE), Neurofilament light chain (NfL) and blood-brain barrier (BBB) proteins [Occludin (OCLN), Zonula occludens 1 (ZO-1), Claudin-5] in blood serum were measured by enzyme-linked immunosorbent assay technique. Biomarker levels in IS patients and SMs were compared using the Mann–Whitney U test. Multivariable logistic regression analysis was used to evaluate the diagnostic performance of biomarkers in combination with the National Institutes of Health Stroke Scale (NIHSS) score. Results: More significant differences in circulating GFAP, NfL, OCLN, ZO-1, and Claudin-5 but not NSE were found in IS patients compared to SMs. A combination of circulating ZO-1, Claudin-5, and OCLN with NIHSS score gives the highest diagnostic accuracy, sensitivity, and specificity. Conclusions: A prediction model with circulating BBB proteins in combination with NIHSS score differentiates between IS patients and SMs.

## 1. Introduction

Stroke is the most common cerebrovascular disease leading to sudden neurological dysfunction caused by a disturbance in the cerebral blood flow due to cerebral ischaemia or haemorrhage [1].

An early and accurate stroke diagnosis is extremely important to reach a good outcome and improve functional recovery. However, early stroke diagnosis can be challenging as it may rely on specialist clinical examination together with expensive and time-consuming neuroimaging techniques, specifically MRI. Whilst non-contrast CT is widely available in emergency departments (ED) and is very sensitive to detecting blood, it is far less sensitive for detecting acute ischaemia and this presents a particular challenge in separating ischaemic stroke (IS) patients from stroke mimics (SMs), commonly functional neurological disorders, migraine, and seizures, which will also often have normal brain CT [2]. Up to 30% of patients presenting to ED with suspected IS using CT will turn out to be SMs [3]. The use of specific circulatory biomarkers related to stroke-associated brain injuries could significantly improve the diagnosis and treatment of stroke patients and post-stroke outcomes and complement the neuroimaging modalities for the diagnosis of stroke [4]. The circulatory biomarkers could be significant in patients with transient neurological symptoms or those who cannot be easily diagnosed by imaging. Moreover, the blood biomarker assessment could be performed during an initial triage, avoiding delays in transporting stroke patients to appropriate care centres with imaging facilities allowing rapid and proper treatments for high-risk patients [4].

Multiple experimental studies have found that stroke damages neurones, astrocytes, and the blood-brain barrier (BBB). These alterations are reflected through proteins released into the blood [5,6]. Recently, different blood-based biomarker panels have been investigated in human stroke [4,7,8,9]. Some biomarkers have shown promising results such as neuron-specific enolase (NSE) [10,11], S100B [12,13], and glial fibrillary acidic protein (GFAP) [14,15].

NSE, a neuron-specific biomarker found mainly in the neuronal cytoplasm [16], has been used in multiple studies to investigate its role in stroke [10]. Neuronal damage assessed by Neurofilament light chain (NfL), a neurone-specific cytoskeletal protein, is reflected in clinical and imaging measurements of illness across different neurological diseases [17].

Numerous studies focused on the role of GFAP in differentiating intracerebral haemorrhage (ICH) and ischaemic stroke suggesting that GFAP may be a potential biomarker for an early prediction of ICH [14,15,18,19,20]. A recent study also reported a positive correlation between serum GFAP level and the National Institutes of Health Stroke Scale (NIHSS) score in acute IS [6].

Occludin (OCLN) is one of the BBB tight-junction (TJ) components. OCLN degradation has been observed in human and animal studies with acute IS leading to BBB breakdown [21,22]. Rapid loss of OCLN protein in the cerebral microvessels has been observed in a rat IS model induced by middle cerebral artery occlusion (MCAO) [23]. Pan et al. (2017) found that OCLN levels in blood increased significantly 4.5 h after MCAO and the increase in blood OCLN levels positively correlated with the extent of BBB damage [24]. Disarrangement of Zonula occludens 1 (ZO-1), another BBB TJ component [25], has been shown to reduce BBB integrity, indicating BBB damage [26]. In contrast, Claudin-5, a key TJ component selectively reduce the permeability to ions [27], did not change significantly after cerebral ischaemia in the rat MCAO model [24,28]. Lasek-Bal et al. (2020) have shown that plasma levels of OCLN and Claudin-5 in acute stroke are correlated with the type and location of stroke [29].

Despite intense efforts in the search for blood-based biomarkers, there is no single circulating biomarker that can be used in hospitals to differentiate between stroke patients and SMs. Most of the studies had a small sample size and very few have compared the biomarker levels in stroke patients specifically with SMs [14,19,30,31,32]. Also, very few studies have assessed biomarker levels in combination with NIHSS scores, which might improve diagnostic accuracy. A recent study by Gaude et al. (2021) has shown that a novel combination of GFAP and D-dimer with NIHSS scores can facilitate the detection of large vessel occlusion [33].

Hence, the main aim of this exploratory study is to assess the ability of blood-based biomarkers (specific to astrocytes and BBB damage) in combination with NIHSS scores to distinguish between IS patients and SMs. This study has been reported according to the Standards for Reporting Diagnostic Accuracy (STARD) guidelines [34].

## 2. Methods

### 2.1. Method Design

This prospective study included suspected IS patients (N = 90) admitted to the stroke unit from the Leeds Teaching Hospitals NHS Trust emergency room at the Leeds General Infirmary (LGI), UK. The study was approved by the Yorkshire and The Humber-Leeds East Research Ethics Committee Patient selection (NHS-REC reference 19/YH/0232, IRAS reference No: 50831). A flow diagram of the study’s design and the patient’s routine clinical pathways is shown in Appendix A.

### 2.2. Patient Selection

Patients included in this study were (a) aged 18+ years and (b) clinically suspected of acute IS when admitted to the hospital, based on a history of acute onset neurological symptoms and examination findings obtained in the emergency rooms.

The exclusion criteria were (a) the patient had capacity but refused consent and (b) the patient lacked capacity and informed consent from the consultee could not be obtained. Patients with rapidly improved symptoms (transient ischaemic attacks) are not routinely admitted to this hospital but referred to outpatient clinics and were not the focus of this study. All patients were recruited randomly between 2021 and 2022.

### 2.3. Recruitment Procedures

Researchers who trained in good clinical practice (GCP) took informed consent from the patients. Patients with acute stroke do not infrequently lose capacity but it is important to include this group to minimise bias. Therefore, consent was also permissible from a relative. All in-patient populations with suspected IS were approached. Patients in the research were not included if it was impossible to obtain a patient informed consent or consultee declaration from a relative. Patient information was retrieved from NHS Trust paper and electronic medical health records (Patient Pathway Manager PPM+). All patient-identifiable data were stored on a password-protected database on the NHS Trust network drive. Data taken offsite to the University for analysis were in a pseudo-anonymised format. The data collected at baseline were: demographic information (age, gender, ethnicity, and occupation), height, weight, waist circumference, blood pressure, comorbidities, vital signs, NIHSS score on admission, and medication on admission. Based on the NIHSS score on arrival, IS patients were classified as mild stroke (NIHSS ≤ 7) and moderate to severe (NIHSS > 7).

### 2.4. Blood Sample Collection

Blood samples from 90 suspected stroke patients (66 were diagnosed as IS patients and 24 as SMs following clinical evaluation) were collected to measure the concentration of biomarkers. For the serum sample, 5 mL of blood was collected in a gel clot active tube (Gold Hemogard closure) and allowed to clot by leaving it undisturbed at room temperature for 30 min. The clot was then removed by centrifuging at 1500× *g* for 10 min at 4 °C. The supernatant (serum) was aliquoted into a 1 mL sterile vial labelled with the patient’s ID, date of blood collection, and time of collection. The samples were flash-frozen using liquid nitrogen and stored at −80 °C until analysis.

### 2.5. Analysis and Measurement of Blood Biomarkers

The ELISA technique was used to quantitatively assess biomarkers following the manufacturer’s protocol. The following ELISA kits were used: human GFAP (Fine test, EH0410, Wuhan, China; sensitivity: 0.188 ng/mL), human ZO-1 (Fine test, EH15434, sensitivity: 0.094 ng/mL), human OCLN (Fine test, EH1674, sensitivity: 18.75 pg/mL), human Claudin-5 (Fine test, EH2839, sensitivity: 0.094 ng/mL), human NSE (R&D Systems, DENL20, Minneapolis, MN, USA; sensitivity: 0.038 ng/mL), human NfL (Abbexa, abx152468, Cambridge, UK; sensitivity: <5.7 pg/mL). All assays were conducted blind to the clinical characteristics of the patients.

### 2.6. Clinical Phenotyping

Routine NHS care was followed in the stroke unit. When the diagnosis of stroke or mimic was uncertain after brain CT, a clinical brain MRI was requested (23% of total participants, IS patients = 9, SMs = 13). All clinical information and radiology were subsequently reviewed by a vascular neurologist blind to the laboratory data. Patients were given a final binary classification of IS or SM by a vascular neurologist using all the available clinical and radiological data, blind to the biomarker data. The clinical and radiological data was reviewed up to the point of the first outpatient follow-up visit from the hospital and this was blind to the biomarker data.

### 2.7. Statistical Analyses

Means and standard deviations were used to summarise numerical variables and frequencies and percentages for categorical variables. The comparison of clinical variables between IS patients and SMs was assessed by student *t*-tests for numerical variables and chi-squared tests for categorical variables. The Mann–Whitney U test was used to compare the blood biomarker levels between IS patients and SMs. Multivariable logistic regression analysis was used to evaluate the diagnostic performance of biomarkers in combination with the NIHSS score to differentiate between IS patients and SMs.

The diagnostic accuracy of the biomarkers was assessed using the area under the receiver operating characteristic curve (AUC), sensitivity, and specificity with a 95% confidence interval (CI). The likelihood ratio (LR) test and Akaike information criterion (AIC) were used to compare nested models of NIHSS score and a combination of biomarkers to select the optimal panel of blood biomarkers (GFAP and BBB proteins) that differentiate IS patients from SMs. The cut-off value was selected by equalising the sensitivity and specificity. Data analysis was performed with Stata 17 MP software. GraphPad Prism 9 software (GraphPad, San Diego, CA, USA) was used for biomarker quantification or analysis and graph presentation. Statistical significance was determined at *p* ≤ 0.05.

## 3. Results

The clinical characteristics of IS patients and SMs are reported in Table 1. In our cohort, we observed significant differences between IS patients and SMs in age, gender, and history of hypertension and type 2 diabetes. There was no difference in admission NIHSS score or onset of symptoms to blood timing (OBT). The frequency of diagnoses in SMs was functional/migraine (10), vestibulopathy (4), demyelination (3), peripheral nerve lesion (3), seizure (2), cardiac failure (1), and head injury (1).

### 3.1. Distinguishing Between Ischaemic Stroke Patients and Stroke Mimics

Table 2 shows the range, mean, and median serum GFAP, NSE, NfL, OCLN, Claudin-5, and ZO-1 concentrations in IS patients and SMs. The levels of serum GFAP were significantly higher in IS patients as compared to SMs with a four-fold increase (*p* < 0.0001, Figure 1A) (N = 90; IS = 66, SMs = 24). No significant difference was observed between the levels of NSE in the serum of IS patients and SMs (*p*  =  0.07, Figure 1B). Serum NfL concentration was 1.33-fold significantly higher in IS patients compared to SMs (*p* < 0.0001, Figure 1C).

The range of serum OCLN, Claudin-5, and ZO-1 in IS patients was significantly higher compared to SMs (*p* < 0.0001, Figure 1D–F).

As the concentrations of NSE in the serum of IS patients were not significantly different compared to SMs, the analysis of NSE was discontinued at N = 63; IS = 46, SMs = 17.

### 3.2. Difference in Biomarker Levels Based on the Severity of Stroke

IS patients were classified as mild stroke (NIHSS ≤ 7) and moderate to severe stroke (NIHSS > 7). Out of 66 IS patients, 56 patients had an NIHSS score ≤ 7 and 10 patients had an NIHSS score > 7. When comparing the relationship between the IS group and SM group and the severity of a stroke, we found that IS patients had significantly higher serum GFAP, NFL, OCLN, Claudin-5, and ZO-1 levels in both groups as compared to SMs (Figure 2A,C–F). IS patients with NIHSS scores ≤ 7 (N = 38) and NIHSS scores > 7 (N = 8) had non-significant changes in serum NSE levels as compared to SMs (N = 17) (*p* = 0.285 and *p* = 0.158, respectively) (Figure 2B).

### 3.3. Combination of Blood Biomarkers with NIHSS

The multivariable logistic regression models that were fitted to determine the optimal panel of biomarkers to differentiate IS patients with SMs are shown in Table 3 and Table 4. Adding biomarkers (GFAP and BBB proteins) to a model with an NIHSS score resulted in a significant reduction in AIC, suggesting a better model fit and an increase in AUC from 50 to greater than 90 showing better discrimination. The combination of selected biomarkers with NIHSS score has significant LR test *p*-values (*p* ≤ 0.05 has been considered statistically significant) showing a significant improvement compared to a model with NIHSS score only.

Amongst the combination model of one biomarker with a stroke severity scale, the NIHSS score + OCLN combination has higher accuracy and sensitivity but lower specificity than other combinations. Adding the GFAP biomarker to this model increases the accuracy and specificity but not sensitivity. Similarly, adding Claudin-5 to the NIHSS score + OCLN combination model does not change any parameter. However, adding Claudin-5 to the NIHSS score + ZO-1 combination model increases the specificity of this model. Thus, we combined ZO-1, Claudin-5, and OCLN with NIHSS score, which gives an accuracy of 89.41 (95% CI: 80.85–95.04), sensitivity of 87.50 (76.85–95.24), and specificity of 95.24 (76.18–99.88), higher among all the combination of three biomarkers with NIHSS score. We also found that testing BBB markers on their own could help diagnose ischaemic stroke with an accuracy of 86.67%.

## 4. Discussion

This novel study for the first time reported a significant difference in circulating BBB TJ proteins (OCLN, ZO-1, and Claudin-5) along with GFAP and NfL but not NSE in IS patients compared to SMs.

The multivariable logistic regression analysis showed that a combination of BBB TJ proteins with NIHSS score gives the highest diagnostic accuracy, sensitivity, and specificity.

Previous studies have shown that GFAP might be utilised as a blood biomarker to distinguish between IS and ICH in the acute stage [35,36]. In the present study, we found that serum GFAP level in IS patients was significantly higher than in SMs. Unlike other studies [37,38,39,40] we did not find that serum NSE concentrations in IS patients were significantly different between IS patients and SMs. Serum NfL was significantly higher in IS patients compared to SMs, confirming previous studies that measured NfL in the blood of IS patients using Simoa [41,42,43,44].

Previous research has shown that the BBB is disrupted during a stroke, resulting in alterations in the concentrations and distributions of Claudin-5, OCLN, ZO-1, and other BBB building blocks [23,27]. Our novel findings showed that these BBB TJ protein levels are significantly higher in IS patients in comparison with SMs. This might reflect the immediate BBB damage, which may not be detected on early CT imaging, hence these biomarkers could potentially have greater sensitivity than imaging strategies routinely available in the emergency department (ED).

A study by Gaude et al. (2021) has shown that a biomarker panel composed of GFAP and D-dimer combined with clinical stroke severity scales can be a valuable tool for the identification of patients with large vessel occlusion (LVO) [33]. Also, a recent study by Jæger et al. (2023) has shown that GFAP combined with the Prehospital stroke scale (PreSS) can identify stroke and stroke subtypes [45]. Herein, we have demonstrated that a biomarker panel of TJ proteins (OCLN, Claudin-5, and ZO-1) combined with a stroke severity scale, NIHSS, can be a good prediction model to differentiate between IS patients and SMs. Whilst multiple studies have shown that NIHSS is a predictor of stroke, this is not sufficiently specific to distinguish cerebral ischaemia from other forms of neuronal injury or dysfunction. The specificity in our study was only 61.9%, which is consistent with a low specificity of NIHSS in the literature. Hand et al. (2016) found that only 73% of patients with an NIHSS ≥ 5 were subsequently confirmed to be a stroke [2]. To our knowledge, no study has established the role of TJ proteins and NIHSS scores in distinguishing between IS patients and SMs. Even for clinicians who are not NIHSS trained, testing the BBB markers in isolation could reliably help with the diagnosis of IS based purely on clinical suspicion from history and/or more limited examination, e.g., the FAST test.

Although blood-based diagnostic tests for cardiac illnesses are often utilised (e.g., troponin I for the diagnosis of myocardial infarction) [46], the use of biomarkers to diagnose stroke is still in its early stages. Blood biomarkers may help physicians make an accurate fast diagnosis of IS, which may be crucial in emergencies where therapies need to be started as soon as possible to enhance patient outcomes. On the other hand, some patients with neurological symptoms that mimic stroke might receive unnecessary interventions. To quickly distinguish and improve the therapeutic care of both IS patients and SMs may also help in the pre-hospital setting to determine whether to transfer patients quickly to a stroke centre. However, in this study, we focused on the ability of selected blood-based biomarkers to distinguish between IS patients and SMs in a population of patients suspected to have IS by ED staff based on clinical triage and brain CT alone.

Potential limitations of this study are that due to COVID-19 restrictions in the ED, we were not able to recruit hyperacute stroke patients and further studies are needed to evaluate the diagnostic performance of these biomarkers in the first few hours after stroke. Mild strokes were over-represented in this study as they were more likely to be recruited due to easier consent. However, this remains an important group to study as they usually reflect lacunar or posterior circulation events which can easily be missed or overdiagnosed in the ED.

Another limitation is that some SMs may have been excluded based on imaging. However, the recruited population is likely to be representative of a population that poses a challenge in the ED and is often referred to stroke teams for further investigations or acute management. Although we controlled for potential confounding factors in multivariable logistic regression, we cannot completely exclude chance effects. Bias is unlikely as the phenotyping was carried out completely blind to the laboratory assays.

## 5. Conclusions

In conclusion, our exploratory data suggest that circulating serum BBB TJ proteins combined with the NIHSS score can differentiate between IS patients and SMs who have been admitted to the hospital from the ED for suspected stroke. Further prospective studies are required to determine the accuracy of these biomarkers in the hyperacute phase of stroke to maximise the clinical impact of a biomarker diagnostic strategy.

## Figures and Tables

**Figure 1 biomolecules-14-01344-f001:**
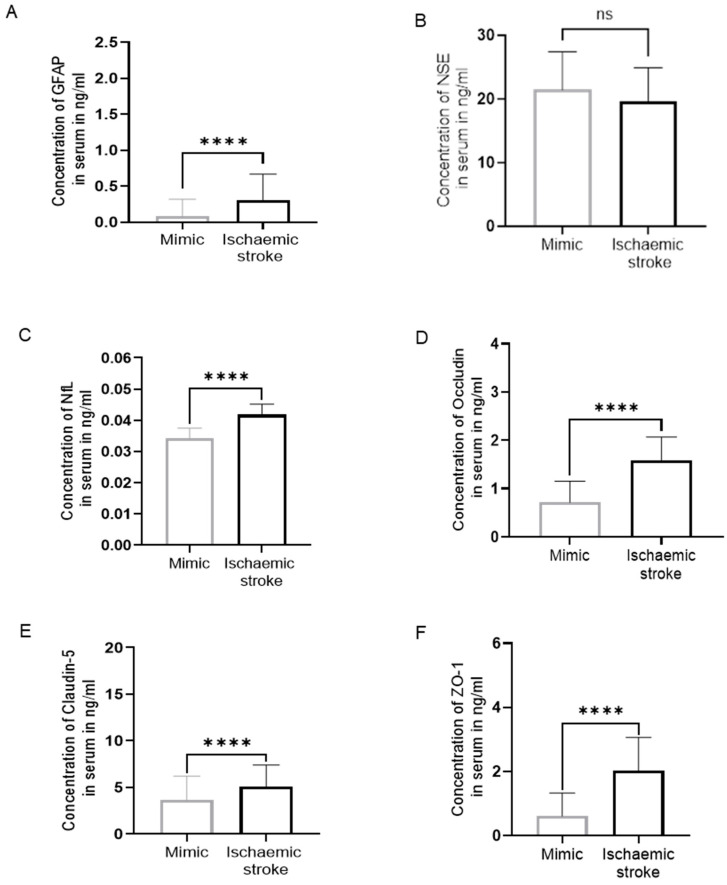
Biomarker levels in serum. (**A**) GFAP, (**B**) NSE, (**C**) NfL, (**D**) Occludin, (**E**) Claudin-5, and (**F**) ZO-1 in ischaemic patients (N = 66) compared to mimics (N = 24). There was a significant increase in the concentration of biomarkers in an ischaemic group compared to mimics except NSE (46 ischaemic patients, 17 mimics). (Mann–Whitney U test, *p* > 0.05: ns, *p* ≤ 0.0001: ****).

**Figure 2 biomolecules-14-01344-f002:**
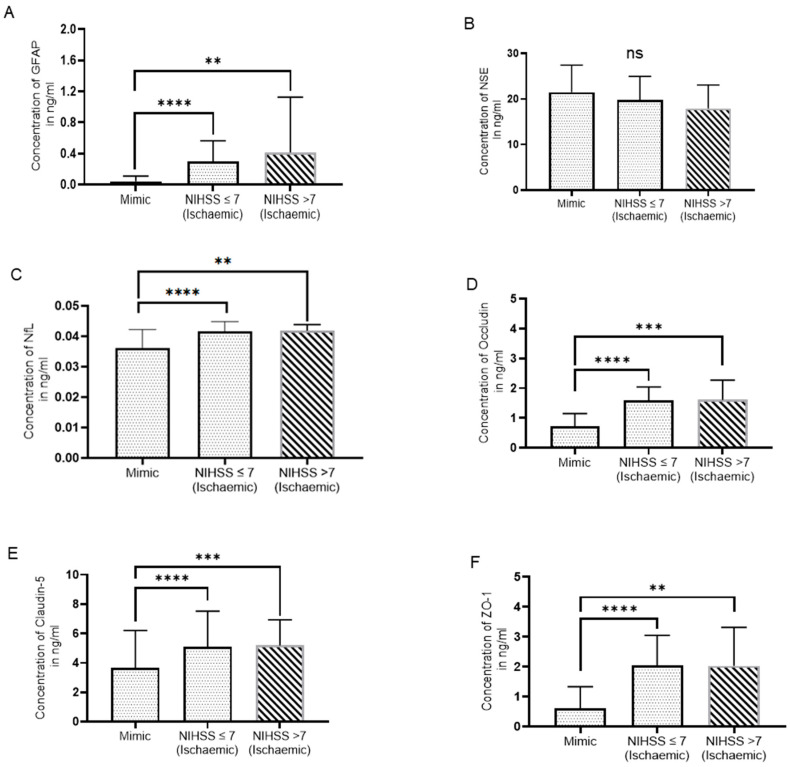
Serum biomarker levels based on NIHSS score. (**A**) GFAP, (**B**) NSE, (**C**) NFL, (**D**) Occludin (**E**) Claudin-5, and (**F**) ZO-1 (Mann–Whitney U test, *p* > 0.05: ns, *p* ≤ 0.01: **, *p* ≤ 0.001: ***, *p* ≤ 0.0001: ****), (NIHSS > 7: N = 10, 8 for NSE, NIHSS ≤ 7: N = 56, 38 for NSE and mimics: N = 24, 17 for NSE).

**Table 1 biomolecules-14-01344-t001:** Univariate analysis of clinical variables in ischaemic stroke patients and stroke mimics.

Clinical Variables	Ischaemic Stroke Patients Mean (SD *)	Stroke Mimics Mean (SD)	*p*-Value
Gender (M/F)	50/16	15/9	<0.001
Age	65.8 (10.9)	54.2 (16.9)	<0.001
Systolic BP ^†^	154.6 (32.1)	149.8 (28.2)	0.53
Diastolic BP	85.8 (18.5)	89.2 (11.4)	0.41
Hypertension (%Yes)	54.5	41.6	<0.001
Diabetes (%Yes)	36.4	33.3	<0.001
APTT ^‡^	30.2 (2.8)	31.2 (3.1)	0.20
Prothrombin time (s)	12.0 (2.4)	11.9 (0.9)	0.90
Platelet count	256.2 (64.1)	244.0 (60.3)	0.42
Red blood count	4.75 (0.51)	4.79 (0.57)	0.74
NIHSS score on admission	3.6 (3.4)	3.0 (2.4)	0.47
OBT ^§^ (days)	3.1 (2.7)	3.3 (2.3)	0.77
Posterior/Anterior Ischaemia	21/45	NA	
Thrombolysis	14	3	

* SD—standard deviation; ^†^ BP—blood pressure; ^‡^ APTT—activated partial thromboplastin time; ^§^ OBT—time difference between stroke onset and blood collection.

**Table 2 biomolecules-14-01344-t002:** The range, mean (SD), and median of serum GFAP, NSE, NfL, OCLN, Claudin-5, and ZO-1 concentrations in the ischaemic stroke group (N = 66 for all biomarkers; N = 46 for NSE) and mimics group (N = 24 for all biomarkers; N = 17 for NSE).

Biomarker		Ischaemic Stroke (ng/mL)	Stroke Mimics (ng/mL)	*p*-Value
GFAP	Mean (SD *)	0.31 (0.36)	0.08 (0.24)	<0.0001
	Minimum	0.02	0.01	
	Maximum	2.41	1.20	
	Median	0.26	0.02	
NSE	Mean (SD)	19.64 (5.24)	21.47 (5.95)	0.07
	Minimum	9.29	12.38	
	Maximum	28.42	27.43	
	Median	20.54	23.66	
NfL	Mean (SD)	0.040 (0.003)	0.030 (0.003)	<0.0001
	Minimum	0.03	0.03	
	Maximum	0.05	0.04	
	Median	0.04	0.03	
OCLN	Mean (SD)	1.59 (0.48)	0.71 (0.44)	<0.0001
	Minimum	0.19	0.25	
	Maximum	3.18	1.66	
	Median	1.64	0.62	
Claudin-5	Mean (SD)	5.09 (2.32)	3.65 (2.55)	<0.0001
	Minimum	0.57	0.74	
	Maximum	17.10	14.99	
	Median	5.51	2.96	
ZO-1	Mean (SD)	2.02 (1.04)	0.61 (0.72)	<0.0001
	Minimum	0.01	0.07	
	Maximum	5.12	3.70	
	Median	2.57	0.42	

* SD—standard deviation.

**Table 3 biomolecules-14-01344-t003:** Different comparison models of biomarkers with NIHSS.

Model	AIC *	AUC ^†^	LR ^‡^, *p*-Value
NIHSS	98.47	52.3 (38.3–66.4)	-
NIHSS + GFAP	81.46	89.5 (80.0–99.0)	19.01, <0.001
NIHSS + ZO-1	66.07	88.7 (78.8–98.6)	34.40, <0.001
NIHSS + OCLN	63.85	90.6 (83.1–98.1)	36.61, <0.001
NIHSS + CLAUDIN-5	90.38	86.1 (75.6–96.7)	10.09, 0.006
NIHSS + GFAP + ZO-1	67.66	89.2 (79.3–99.0)	34.81, <0.001
NIHSS + GFAP + OCLN	64.84	90.1 (80.6–99.6)	37.62, <0.001
NIHSS + GFAP + CLAUDIN-5	81.53	91.7 (82.1–100)	20.94, <0.001
NIHSS + ZO-1 + OCLN	63.15	90.3 (81.1–99.6)	39.32, <0.001
NIHSS + ZO-1 + CLAUDIN-5	66.64	90.6 (83.1–98.1)	35.83, <0.001
NIHSS + OCLN + CLAUDIN-5	65.62	91.4 (84.9–97.8)	36.85, <0.001
NIHSS + GFAP + ZO-1 + OCLN	65.07	90.3 (81.1–99.6)	39.40, <0.001
NIHSS + GFAP + ZO-1 + CLAUDIN-5	68.41	90.5 (81.8–99.1)	36.06, <0.001
NIHSS + GFAP + OCLN + CLAUDIN-5	65.05	91.4 (84.5–98.1)	39.42, <0.001
NIHSS + ZO-1 + OCLN + CLAUDIN-5	62.54	92.1 (86.1–98.1)	41.92, <0.001
NIHSS + GFAP + ZO-1 + OCLN + CLAUDIN-5	64.36	92.2 (86.3–98.0)	42.11, <0.001
ZO-1 + OCLN + CLAUDIN-5	65.85	93.0 (87.7–98.3)	-

* AIC—Akaike information criterion; ^†^ AUC—area under the receiver operating characteristic curve presented with 95% confidence intervals; ^‡^ LR—likelihood ratio.

**Table 4 biomolecules-14-01344-t004:** Accuracy, sensitivity, and specificity values of different combinations of biomarkers with NIHSS score.

Model	Accuracy	Sensitivity	Specificity
NIHSS	47.06 (36.13–58.19)	43.19 (29.94–55.18)	61.90 (38.44–81.89)
NIHSS + GFAP	83.53 (73.91–90.69)	79.69 (67.77–88.72)	95.24 (76.18–99.88)
NIHSS + ZO-1	85.88 (76.64–92.49)	83.81 (71.32–91.10)	95.24 (76.18–99.88)
NIHSS + OCLN	87.06 (78.02–93.36)	89.06 (78.75–95.49)	80.95 (58.09–94.55)
NIHSS + CLAUDIN-5	77.65 (67.31–85.97)	71.88 (59.24–82.40)	95.24 (76.18–99.88)
NIHSS + GFAP + ZO-1	87.06 (78.02–93.36)	84.38 (73.14–92.24)	95.24 (76.18–99.88)
NIHSS + GFAP + OCLN	89.41 (80.85–95.04)	89.06 (78.75–95.49)	90.48 (69.62–98.83)
NIHSS + GFAP + CLAUDIN-5	84.71 (75.27–91.60)	81.25 (69.54–89.92)	95.24 (76.18–99.88)
NIHSS + ZO-1 + OCLN	88.24 (79.43–94.21)	85.94 (74.98–93.36)	95.24 (76.18–99.88)
NIHSS + ZO-1 + CLAUDIN-5	85.88 (76.64–92.49)	82.81 (71.32–91.10)	95.24 (76.18–99.88)
NIHSS + OCLN + CLAUDIN-5	87.06 (78.02–93.36)	89.06 (78.75–95.49)	80.95 (58.09–94.55)
NIHSS + GFAP + ZO-1 + OCLN	88.24 (79.43–94.21)	85.94 (74.98–93.36)	95.24 (76.18–99.88)
NIHSS + GFAP + ZO-1 + CLAUDIN-5	85.88 (76.64–92.49)	82.81 (71.32–91.10)	95.24 (76.18–99.88)
NIHSS + GFAP + OCLN + CLAUDIN-5	88.24 (79.43–94.21)	87.50 (76.85–94.45)	90.48 (69.62–98.83)
NIHSS + ZO-1 + OCLN + CLAUDIN-5	89.41 (80.85–95.04)	87.50 (76.85–94.45)	95.24 (76.18–99.88)
NIHSS + GFAP + ZO-1 + OCLN + CLAUDIN-5	89.41 (80.85–95.04)	87.50 (76.85–94.45)	95.24 (76.18–99.88)
ZO-1 + OCLN + CLAUDIN-5	86.67 (77.87–92.92)	83.33 (72.13–95.38)	95.83 (78.88–99.89)

All diagnostic values are presented with 95% confidence intervals.

## Data Availability

The data presented in this study are available on request from the corresponding author due to privacy or ethical reasons.

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
