# Peer review of "Circulating Blood-Brain Barrier Proteins for Differentiating Ischaemic Stroke Patients from Stroke Mimics"

_biomolecules, 2024, doi:10.3390/biom14111344_

Round 1
Reviewer 1 Report
Comments and Suggestions for Authors
The paper is well writing.
Comments:
The definition of suspected acute IS should be clarified. Is it defined by time of neurologic deficit, or in combination with imaging, or others? How to exclude TIA? The OBT was short, did the time from syptom onset to recruitment was requested, and how?
Author Response
"Please see the attachment"

Reviewer 2 Report
Comments and Suggestions for Authors
The manuscript entitled “Circulating blood-brain barrier proteins for differentiating ischaemic stroke patients from stroke mimics,” utilizes human serum, from suspected ischemic stroke patients in the emergency department, to measure protein markers of neurons, astrocytes, and blood brain barrier health relative to the NIHSS. The use of blood biomarkers in the ED to assist in early diagnosis of ischemic stroke before imaging is available is extremely valuable and needed. Additional sample information is needed in methodology for conclusions. The authors may want to consider the following major and minor comments:
Major:
1. The authors may want to consider adding additional background to the introduction as the rational for picking the biomarkers was extremely limited. The authors may also want to consider additional background in humans, not only rodents, for many of the BBB blood markers.
2. The background utilizes studies from hemorrhagic and ischemic stroke, but then only uses patients of ischemic stroke. The authors may want to elaborate on how the two types of stroke present and why IS was used.
3. The authors state few reports have assessed biomarkers in combination with the NIHSS. Please expand on this statement for clarity as I believe many IS studies incorporate NIHSS. Is there data indicating the specificity of the NIHSS for IS?
4. Additional details regarding time of blood draw to symptom onset and time in the ED are needed, as well as, timing of the NIHSS assessment in regards to the blood draw to accurately draw conclusions from the data. Did any of the IS subjects have TPA or thrombectomies?
5. Please provide data as to the number of each subtype (i.e., head injury, migraine, etc) in the SM group. What was the NIHSS numbers for IS and SM? Could the severity of the suspected stroke indicate injury rather than ischemic stroke?
Minor:
6. Did the authors model the severity of the stroke (NIHSS) with the multivariable logistic regression models for IS compared to SM?
7. Can the authors please elaborate as to why NfL was not included in tables 3 and 4?
8. Since the samples were stored in 1 mL aliquots, did each sample undergo multiple freeze thaws? If so, how many?
9. The order of the proteins in the graphs is not the same as the table.
Author Response
"Please see the attachment"

Round 2
Reviewer 2 Report
Comments and Suggestions for Authors
My only comment is that references are needed for the new statements in the introduction.
Author Response
Comment 1: My only comment is that references are needed for the new statements in the introduction.
Response: The relevant references have been added for the new statements in the introduction.